# Therapeutic Approaches to Dysautonomia in Childhood, with a Special Focus on Long COVID

**DOI:** 10.3390/children10020316

**Published:** 2023-02-07

**Authors:** Reiner Buchhorn

**Affiliations:** Praxis for Pediatrics, Pediatric Cardiology and Adults with Congenital Heart Diasease, Am Bahnhof 1, 74670 Forchtenberg, Germany; buchrein@gmail.com; Tel.: +49-794-7919-081; Fax: +49-947-919-091

**Keywords:** autonomic nervous system, heart rate variability, COVID-19, anorexia nervosa, autoimmunity, omega-3 fatty acids, betablocker, ivabradine, inappropriate sinus tachycardia, postural orthostatic tachycardia syndrome

## Abstract

Background: Dysautonomia seems to be important for the pathophysiology of psychosomatic diseases and, more recently, for long COVID. This concept may explain the clinical symptoms and could help open new therapeutic approaches. Methods: We compared our data from an analysis of heart rate variability (HRV) in an active standing test in 28 adolescents who had developed an inappropriate sinus tachycardia (IST, *n* = 13) or postural orthostatic tachycardia syndrome (POTS, *n* = 15) after contracting COVID-19 disease and/or vaccination with 64 adolescents from our database who developed dysautonomia due to psychosomatic diseases prior to the COVID-19 pandemic. We prove the effects of our treatment: omega-3 fatty acid supplementation (O3-FA, *n* = 18) in addition to propranolol (low dose, up to 20-20-0 mg, *n* = 32) or ivabradine 5-5-0 mg (*n* = 17) on heart rate regulation and heart rate variability (HRV). Results: The HRV data were not different between the adolescents with SARS-CoV-2-related disorders and the adolescents with dysautonomia prior to the pandemic. The heart rate increases in children with POTS while standing were significantly lower after low-dose propranolol (27.2 ± 17.4 bpm***), ivabradine (23.6 ± 8.12 bpm*), and O-3-FA (25.6 ± 8.4 bpm*). The heart rate in children with IST while lying/standing was significantly lower after propranolol (81.6 ± 10.1 bpm**/101.8 ± 18.8***), ivabradine (84.2 ± 8.4 bpm***/105.4 ± 14.6**), and O-3-FA (88.6 ± 7.9 bpm*/112.1/14.9*). Conclusions: The HRV data of adolescents with dysautonomia after COVID-19 disease/vaccination are not significantly different from a historical control of adolescents with dysautonomia due to psychosomatic diseases prior to the pandemic. Low-dose propranolol > ivabradine > omega-3 fatty acids significantly decrease elevated heart rates in patients with IST and the heart rate increases in patients with POTS and may be beneficial in these children with dysautonomia.

## 1. Introduction

Dysautonomia is now recognized as an important player in the pathophysiology of psychosomatic diseases like Myalgic encephalomyelitis/chronic fatigue syndrome, often following an acute viral or bacterial infection. This concept has been neglected in pediatrics for a long time [1] but is currently getting a lot of attention in the context of the COVID-19 pandemic and the so-called long COVID syndrome.

We investigate the impact of the heart–brain-axis on emotional regulation as a window of opportunity for new therapeutic approaches within the last 25 years. We currently observe a further increase of such nonspecific symptoms in children and adults during the COVID-19 pandemic, as was shown in the three-wave longitudinal COPSY study [2], that may be related to the impact of SARS-CoV-2 infections on the autonomic nervous system [3,4]. After contracting the acute disease, the ongoing health problems of COVID-19 are now being identified, with the most common being fatigue, postexertional malaise, cognitive dysfunction, memory loss, headaches, orthostatic intolerance, sleep difficulty, and shortness of breath. These ongoing health problems have been termed the postacute sequalae of SARS-CoV-2 infection (PASC) [5]. At least 65 million individuals worldwide are estimated to have PASC [6]. The impact of SARS-CoV-2 vaccination on PASC is controversial. Recent findings suggest that vaccination before infection confers only partial protection against PASC. In addition, new data have been published that show SARS-CoV-2 vaccination may cause PASC-like health problems [7] and POTS [8].

The current publication is a retrospective analysis of our routine clinical data that compare the heart rate analysis from 28 adolescents who developed an inappropriate sinus tachycardia (IST, *n* = 13) or a postural orthostatic tachycardia syndrome (POTS, *n* = 15) after contracting the COVID-19 disease and/or vaccination with 64 adolescents from our database who developed dysautonomia due to psychosomatic diseases prior to the COVID-19 pandemic (IST, *n* = 20 and POTS = 44). The definitions of POTS and IST depend on the current expert consensus statement on the diagnosis and treatment of postural tachycardia syndrome, inappropriate sinus tachycardia, and vasovagal syncope [9]. However, while using an active standing test with continuous heart rate analysis, we use a methodological modification using average heart rates while 5 min of lying down and 5 min of standing with a proven cut-off heart rate increase of 35 bpm for POTS diagnosis in adolescents, as was recently published [10]. We diagnosed an inappropriate sinus tachycardia if the lying heart rate was above 90 bpm.

After checking the comparability of these data in relation to the pandemic, we analyzed the impact of therapy using low-dose propranolol or ivabradine in addition to supplementation with omega-3 fatty acids on heart rates in an active standing test in the whole group of children with dysautonomia.

## 2. Materials and Methods

### 2.1. Patients

As shown in Figure 1, we retrospectively analyzed the data of 92 consecutive children (28 patients COVID-19 related and 64 patients prior to the pandemic) with the diagnosis of an inappropriate sinus tachycardia (IST, *n* = 33, age 15.0 ± 2.6 years) or a postural orthostatic tachycardia syndrome (POTS, *n* = 59, age 14.8 ± 2.4 years) who had had an active standing test within the last five years, conducted in the pediatric department of the Caritas Hospital in Bad Mergentheim and my private praxis in Forchtenberg. The patients were presented in the outpatient clinic with the following symptoms: syncope, dizziness, gastrointestinal dysmotility, eating disorders, headaches, Raynaud’s phenomena, and palpitations. However, the most common long-term symptom was fatigue or weakness. After the diagnosis of a postural orthostatic tachycardia syndrome or an inappropriate sinus tachycardia in the active standing test, the group was divided into patients who presented prior to the pandemic and those who were related to a PCR-proven COVID-19 disease (*n* = 13) or SARS-CoV-2 mRNA vaccination (*n* = 6, Comirnaty™; BioNTech/Pfizer, Berlin, Germany). A further 9 patients had a breakthrough infection after SARS-CoV-2 mRNA vaccination. In summary, 15 out of the 28 patients with COVID-19 related dysautonomia were vaccinated.

The methodology of the active standing test and the healthy control data of 47 children with a comparable mean age of 14.2 ± 3.8 years are published [10]. According to this methodology, postural orthostatic tachycardia syndrome is defined by an increase in average heart rate of more than 35 bpm while standing, and for inappropriate sinus tachycardia, a heart rate ≥ 90 bpm in the lying position. The data are stored in a database by the company’s software (HRV Scanner™, BioSign GmbH, Ottenhofen, Germany). Children with acute and chronic somatic diseases were excluded.

### 2.2. HRV Analysis

The definitions and interpretation of HRV parameters were standardized according to the Task Force Guidelines [11].

HRV analysis was performed using the HRV Scanner™ (BioSign GmbH, Germany) for 5 min in the supine position and a further 5 min in active standing. Blood pressure measurement was carried out in the supine position.

#### 2.2.1. Time Domain HRV

Average heart rates in beats per minute = mean heart rates of each 5 min interval;rMSSD in milliseconds = root mean square of differences between successive NN intervals; this parameter reflects parasympathetic influence.

#### 2.2.2. Stress Index

The stress index is becoming increasingly popular because it reacts sensitively to shifts in the vegetative balance between the sympathetic and parasympathetic nerves.
$$Stres\sin dex=\frac{Amo}{2\times Mo\times MxDMn}$$

Amo = number of RR intervals corresponding to the mode as a percentage of the total number of all readings; Mo = modal value, most common value of the RR intervals; MxDMn = variability width, difference between the maximum and minimum RR intervals.

#### 2.2.3. Frequency Domain HRV

The Fourier transformation concentrated the HRV-signals into three different frequency bands: Very low-frequency power (VLF = 0.00–0.04 Hz) in ms^2^;Low-frequency power (LF = 0.04–0.15 Hz) in ms^2^;High-frequency power (HF = 0.15–0.4 Hz) in ms^2^;LF/HF ratio;Total power (TP) in ms^2^.

The interpretation of the Fourier transformation is controversial, but the high-frequency power clearly reflects respiratory sinus arrhythmia, which depends on vagus activity. The total power measures the total variance in HRV.

### 2.3. Pharmacotherapy and Nutritional Supplementation

For the current analysis, we investigated the effect of a new therapy with low-dose propranolol and ivabradine in children with POTS or IST in the active standing test: propranolol: 10-10-0, up to 20-20-0 mg (*n* = 32), ivabradine: 5-5-0 mg (*n* = 18). This therapy was based upon a consensus statement of the Heart Rhythm Society published in 2015 [9].

We further investigated the impact of omega-3 fatty acid supplementation (O3-FA, *n* = 18) on heart rate in the active standing test. As recently published, we introduced O3-FA supplementation in children with inappropriate sinus tachycardia [12] after showing a significant reduction in the mean heart rate in 24 h ECGs in accordance with a recent meta-analysis [13]. Patients usually purchased products based upon 1–2 g fish oil per day rate from a retail store. The adolescents received at least 800 mg eicosapentaenoic acid (EPA) and docosahexaenoic acid (DHA) per day.

In the first visit, children with dysautonomia were provided lifestyle advice, including increased fluid and salt intake, low-dose exercise, and omega-3 fatty acid supplementation. If these lifestyle interventions were not successful, we introduced pharmacotherapy, first with low-dose propranolol and second with ivabradine if the propranolol did not improve the clinical symptoms. Omega-3 fatty acid supplementation was not stopped during pharmacotherapy.

### 2.4. Statistics

All analyses were performed using IBM SPSS Statistics software (IBM Corp. IBM SPSS Statistics for Windows, Version 27.0, Armonk, NY, USA). For descriptive statistics, data were expressed as mean ± standard deviation. The study population was divided into two diagnosis groups (POTS, *n* = 59 and IST, *n* = 33). These diagnosis groups were subdivided according to COVID-19/vaccination-related patients, patients prior to the pandemic, and one healthy control group that was published and measured prior to the pandemic. An unpaired t-test was used to compare the differences between each patient group (Table 1). Significant group differences were anticipated if the *p*-value was <0.05. For the analysis of the impact of low-dose propranolol, ivabradine, and omega-3 fatty acid supplementation, we used a paired t-test at baseline in comparison to an active standing test after the introduction of these therapies (Table 2).

## 3. Results

As shown in Table 1, the age, height, and body weight are not significantly different within the patient groups and the healthy control. There are highly significant differences between the patient groups and the healthy control group based upon the definitions of a heart rate increase while standing of ≥35 bpm in children with POTS and a lying heart rate of ≥ 90 bpm in children with IST that are not given in this analysis. The statistical results in Table 1 display the unpaired t-tests between the POTS and IST patients prior to the pandemic compared to children with dysautonomia due to PASC and/or vaccination.

The blood pressures are normal on average. However, compared to our historical data, the children with PASC showed significantly higher systolic/diastolic blood pressures if they developed inappropriate sinus tachycardia and lowered systolic blood pressures if they developed postural orthostatic tachycardia. Myocardial function, measured by the fractional shortening and the left ventricular performance index, was normal in all patients, with a very low myocardial performance index that indicates good myocardial function in the children with postural orthostatic tachycardia in the COVID-19-related group.

If HRV in the active standing test of the children with dysautonomia is not different after COVID-19 disease/vaccination compared to a historical control: children with dysautonomia due to psychosomatic diseases prior to the pandemic, then we can prove the effect of omega-3 fatty acid supplementation and pharmacotherapy with low-dose propranolol and ivabradine in the whole group, as displayed in Table 2. The heart rate increases in children with POTS while standing were significantly lower after low-dose propranolol (27.2 ± 17.4 bpm***), ivabradine (23.6 ± 8.12 bpm*), and omega-3 fatty acids (25.6 ± 8.4 bpm*). All these therapeutics have nearly no effect on the lying heart rate, which was normal on average, but they had a significant effect on the standing heart rate, which was elevated on average. This effect was accompanied by an increase in the low vagus activity indicated by the parameter RMSSD and a decrease in the elevated stress index while standing only in the low-dose propranolol group.

The elevated heart rates in the children with IST while lying/standing were significantly lower after propranolol (81.6 ± 10.1 bpm**/101.8 ± 18.8***), ivabradine (84.2 ± 8.4 bpm***/105.4 ± 14.6**), and O-3-FA (88.6 ± 7.9 bpm*/112.1/14.9*). Again, this effect was accompanied by an increase in low vagus activity indicated by the parameter RMSSD and a decrease in the elevated stress index while standing and lying only in the low-dose propranolol group.

The baseline heart rates of the children with dysautonomia and the effect of the three therapeutic approaches on heart rate while lying and standing for the POTS group are shown in Figure 2, and Figure 3 shows the IST group.

## 4. Discussion

PASC or long COVID seems to be a new disease with a known origin—a SARS-CoV-2 infection and/or probably vaccination—but with unknown pathophysiology and therapy. Early in the pandemic, we suspect that dysautonomia was a possible cause of the complaints, as we know the relationship between chronic fatigue syndrome and other infections like Epstein–Barr virus infections (Figure 4) [14]. Our therapeutic approach—as analyzed in this publication—is based upon our experience with these children with dysautonomia. We decided on using a continuous heart rate analysis for 5 min of lying down and standing up to eliminate the (in part) high variations in heart rate due to respiratory sinus arrhythmia in the young (Figure 4a). Based upon this modification, we found slightly other normal values in adolescents, as was recently published [10], and used average heart rate increases of more than 35 bpm for the diagnosis of a postural orthostatic tachycardia syndrome. The 3D plot of the spectral analysis of heart rate variability clearly shows the collapse of the complete spectrum of HRV while standing (Figure 4b) in a 13-yea-old boy with chronic fatigue after an Epstein–Barr virus infection. If most patients suffer from unspecific clinical symptoms without a correlate in the routine diagnostics, the patients already benefit from such objectification of their complaints.

We observed similar changes in heart rate regulation in most children with long COVID. To date, we found 28 adolescents who have developed postural orthostatic tachycardia syndrome (*n* = 15) or inappropriate sinus tachycardia (*n* = 13) after COVID-19 disease and/or vaccination (15 out of 28 patients with COVID-19-related dysautonomia were vaccinated). The high level of suffering of the children prompted us to initiate therapy in accordance with the current expert consensus statement on the diagnosis and treatment of postural orthostatic tachycardia syndrome, inappropriate sinus tachycardia, and vasovagal syncope [9]. Furthermore, together with the established lifestyle advice for children with dysautonomia, we introduced omega-3 fatty acid supplementation. As recently published in PlosOne [12], we found a significant heart rate decrease in the 24 h Holter ECGs of adolescents with inappropriate sinus tachycardia.

For a better understanding of our therapeutic approaches, we recommend the current reviews on inappropriate sinus tachycardia [15] and postural orthostatic tachycardia syndrome [16], which updated our knowledge with respect to the auto-antibodies after COVID-19 infection that interferes with the receptors of the autonomic nervous system. We performed auto-antibody analysis in a research laboratory [17] and found different patterns of elevated auto-antibodies against G-protein-coupled receptors in children with long COVID and also after SARS-CoV-2 vaccination [18]. However, the auto-antibody concentrations against these receptors were not significantly different between the controls and the patients with POTS in a recent analysis [19].

In the current retrospective analysis, we address the following questions for the transfer of our established treatment to PASC—a newly occurring disease:(1)Is PASC a comparable disease of the autonomic nervous system, similar to what we know from the treatment of psychosomatic diseases prior to the pandemic?(2)Can we objectively measure an effect on heart rate regulation for our most commonly used treatments (low-dose propranolol, ivabradine, and omega-3 fatty acid supplementation)?

As shown in Table 1, our analysis of heart rate variability in the active standing test in children with dysautonomia after COVID-19 disease/vaccination is not significantly different from the historical controls of children with dysautonomia due to psychosomatic diseases prior to the pandemic. However, the blood pressures in the children with PASC who develop inappropriate sinus tachycardia are significantly higher on average compared to the historical control. These data are in good accordance with our recent publication, which shows elevated diastolic blood pressures in all children with PASC compared to healthy controls [20]. Moreover, in a linear regression analysis of our full cohort of 479 adolescents, we found that the systolic blood pressure percentile depends primarily on the body mass index and mean 24 h heart rate [12]. Based on these data, children with PASC who develop inappropriate sinus tachycardia may have an enhanced cardiovascular risk, as shown by the Swedish register data [21]. In contrast, children with PASC who develop postural orthostatic tachycardia syndrome have normal heart rates in the lying position (69.2 ± 13.2 bpm) and normal systolic blood pressures (110.3/69.5 mmHg) on average compared to the historical controls (72.5 ± 10.9 bpm, 116.7/63.5 mmHg) and the healthy control (73.6 ± 12.5 bpm, 114.5/61.7 mmHg). One reason for this observation may be that, in our group of children with PASC who developed postural orthostatic tachycardia syndrome, many of the children were active athletes with low resting heart rates and probably low cardiovascular risk.

As shown in Figure 2 and Table 2, our most common treatments significantly reduced the heart rate increases in the active standing test in those patients with postural orthostatic tachycardia syndrome (low-dose propranolol: from 42.2 ± 15.9 to 27.2 ± 17.4 bpm***, ivabradine from 37.5 ± 16.4 to 23.6 ± 8.12 bpm*, omega-3 fatty acid supplementation from 44.0 ± 11.9 to 25.6 ± 8.4 bpm*), with the most significant effect on heart rate while standing. In patients with inappropriate sinus tachycardia, these therapeutics decrease heart rates while lying (propranolol: from 102.7 ± 20.8 to 81.6 ± 10.1 bpm**, ivabradine: from 103.3 ± 12.2 to 84.2 ± 8.4 bpm***, omega-3 fatty acids: from 96.4 ± 8.6 to 88.6 ± 7.9 bpm*) and standing (propranolol: from 132.1 ± 16.5 to 101.0 ± 18.8 bpm***, ivabradine: from 128.6 ± 12.2 to 105.4 ± 14.6 bpm**, omega-3 fatty acids: from 121.5 ± 11.9 to 112.1 ± 14.9 bpm*) (Figure 3, Table 2). 

Our minimum therapeutic goals to reduce elevated heart rates and heart rate increases were achieved by all three substances. Low-dose propranolol was the most effective treatment to reduce the elevated heart rates, with a significant effect on heart rate variability as indicated by the vagus parameter RMSSD and the stress index. Ivabradine and omega-3 fatty acid supplementation had a minor and insignificant effect on heart rate variability in the current analysis. However, based on an unsatisfactory clinical effect, we had to change low-dose propranolol to ivabradine in 17 patients in the whole group. Moreover, nearly all patients had baseline therapy with omega-3 fatty acid supplementation that was not stopped during pharmacotherapy.

In the current analysis, our patients with PASC who had developed inappropriate sinus tachycardia were treated with low-dose propranolol (*n* = 3), ivabradine (*n* = 1) or omega-3 fatty acid supplementation alone (*n* = 3). The patients with PASC who had developed postural orthostatic tachycardia syndrome received low-dose propranolol (*n* = 5) or ivabradine (*n* = 2) together with omega-3 fatty acid supplementation.

It has been shown that higher heart rates and heart rate increases while standing may indicate more emotional stress [22]. Cognitive dysfunction is common in patients with postural orthostatic tachycardia syndrome [23] and may explain school difficulties in adolescents. Moreover, the treatment of dysautonomia in patients with postural orthostatic tachycardia syndrome may reduce symptom burdens [24]. In these patients, low-dose propranolol had a proven, beneficial effect on maximum exercise capacity measured 1 h after medication [25], and higher-dose propranolol did not further improve this and may even worsen the symptoms [26]. Ivabradine has a proven beneficial effect on the clinical symptoms of patients with inappropriate sinus tachycardia and completely eliminated them in approximately half of the patients [27]. These studies that were conducted a relatively long time ago have received little attention. However, they represent great hope to millions of patients who currently suffer from long COVID.

However, our data, which are based on a retrospective analysis of routine clinical data, have some limitations: To date, some patients with PASC who are treated for dysautonomia do not have a second active standing test as a therapy control. Based on this analysis, our therapeutic approach, which is based on heart rate analysis, seems to be justified. However, our retrospective data must be proven in a prospective randomized trial with additional clinical endpoints. Moreover, we are not able to prove the effect of SARS CoV vaccination on PASC. A total of 15 out of 28 patients with COVID-19-related dysautonomia were vaccinated, which agrees well with the current vaccination rate of 67% for this age group in Baden-Württemberg, Germany.

In conclusion, our HRV data of adolescents with dysautonomia after COVID-19 disease/vaccination are not significantly different from a historical control of adolescents with dysautonomia due to psychosomatic diseases prior to the pandemic. Low-dose propranolol > ivabradine > omega-3 fatty acids significantly decrease elevated heart rates in patients with IST and the heart rate increases in patients with POTS and may be beneficial in these children with dysautonomia.

## Figures and Tables

**Figure 1 children-10-00316-f001:**
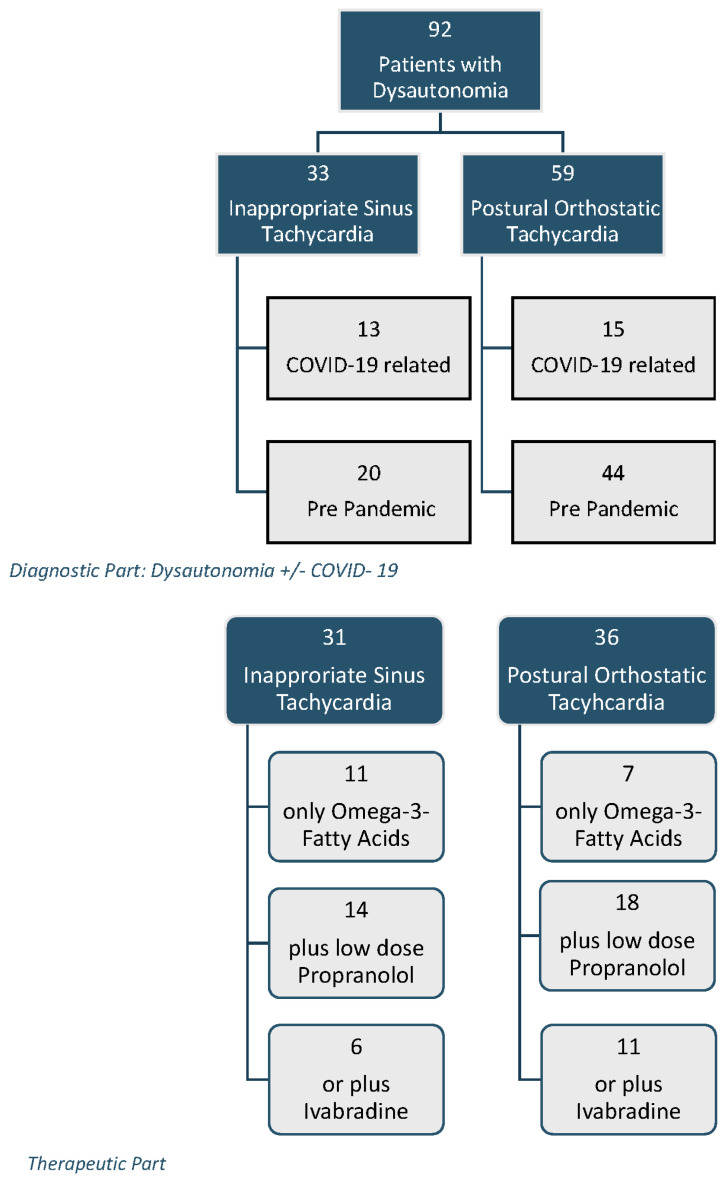
Number of patients and study groups.

**Figure 2 children-10-00316-f002:**
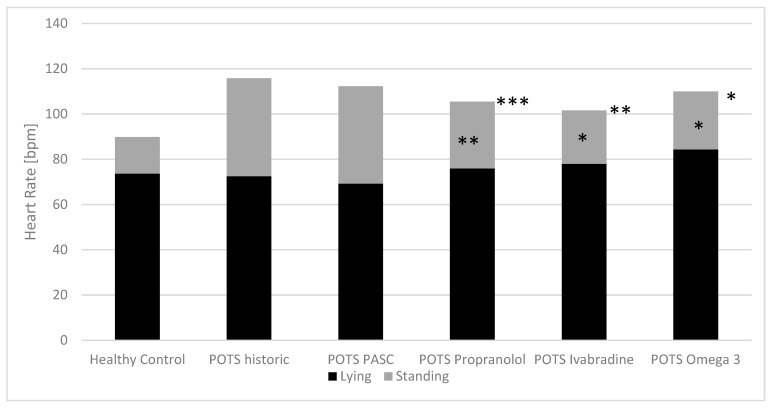
Baseline heart rates in the active standing test and the effect of treatment in children with postural orthostatic tachycardia syndrome. PASC: postacute sequalae of SARS-CoV-2 infection/vaccination; POTS: postural orthostatic tachycardia syndrome. Paired t-test between baseline and treatment: * *p*-value < 0.05; b ** *p*-value < 0.01; *** *p*-value < 0.001.

**Figure 3 children-10-00316-f003:**
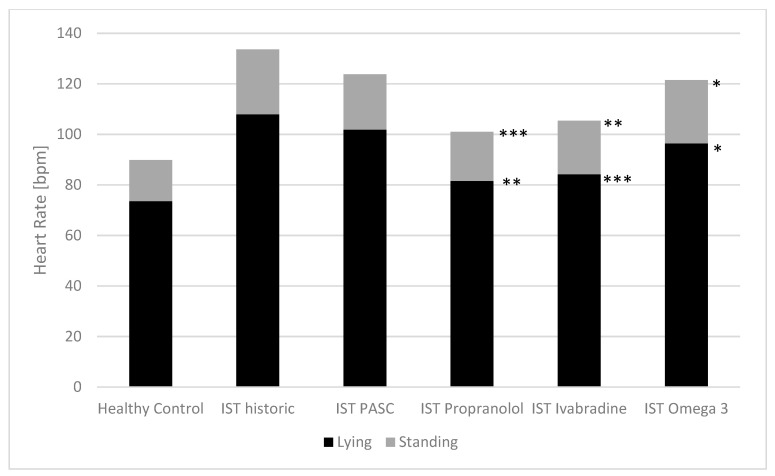
Baseline heart rates in the active standing test and the effect of treatment in children with inappropriate sinus tachycardia. PASC: Postacute sequalae of SARS-CoV-2 infection/vaccination; IST: inappropriate sinus tachycardia. Paired t-test between baseline and treatment: * *p*-value < 0.05; b ** *p*-value < 0.01; *** *p*-value < 0.001.

**Figure 4 children-10-00316-f004:**
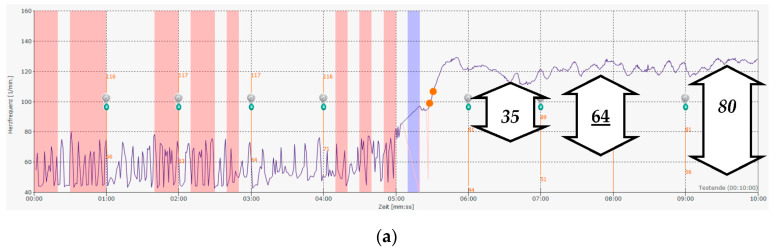
A 13-year-old boy with chronic fatigue after Epstein–Barr virus infection and postural orthostatic tachycardia syndrome: (**a**) The original heart rate raw data: the software calculates an average increase in heart rate of 64 bpm (POTS cut-off is 35 bpm), as used in the current analysis. Frequently used point measurements (could be over 80 bpm—or underestimated at 35 bpm) for the heart rate increases due to the pronounced respiratory sinus arrhythmia in many adolescents. (**b**) The monitoring of heart rate variability with the power spectral analysis clearly demonstrates the nearly complete collapse of heart rate variability while standing, which affects almost all frequencies.

**Table 1 children-10-00316-t001:** Measurements during active standing test in children with dysautonomia.

	Healthy Control	Inappropriate Sinus Tachycardia		Postural Orthostatic Tachycardia	
		Prior to Pandemic	PASC/Post Vaccination	Prior to Pandemic	PASC/Post Vaccination
**Patients**	47	20	13	44	15
Age [Years]	14.2 ± 3.8	14.7 ± 2.5	15.4 ± 2.9	14.9 ± 2.2	14.4 ± 2.7
Height [cm]	160.1 ± 14.2	160.9 ± 9.8	162.1 ± 8.3	167 ± 10	165 ± 12
Weight [kg]	52.6 ± 14.3	48.2 ± 7.7	53.0 ± 12.3	55.4 ± 11.6	54.5 ± 12.7
Sys. BP [mmHg]	114.5 ± 9.2	119.4 ± 9.5	129.1 ± 7.3 **	116.7 ± 9.3	110.3 ± 2.8 *
Diast. BP [mmHg]	61.7 ± 11.2	73.6 ± 9.5	80.4 ± 8.6 *	63.5 ± 11.6	69.5 ± 6.0
FS [%]		38.9 ± 6.2	37.0 ± 3.6	44.9 ± 5.3	37.8 ± 4.8 *
LVIMP		0.13 ± 0.1	0.20 ± 0.1	0.22 ± 0.08	0.1 ± 0.07 **
HR Increase	16.2 ± 7.1	25.7 ± 14.5	21.9 ± 12.6	43.3 ± 8.7	43.0 ± 5.8
Lying HR	73.6 ± 12.5	107.9 ± 16.2	101.9 ± 9.2	72.5 ± 10.9	69.2 ± 13.2
Standing HR	89.8 ± 13.2	133.6 ± 16.4	123.8 ± 12.6 *	115.6 ± 15.3	112.3 ± 13.4
rMSSD Lying	85.1 ± 56.2	19.6 ± 15.2	22.8 ± 14.1	63.4 ± 41.4	88.7 ± 46.0
rMSSD Standing	40.4 ± 22.7	12.5 ± 15.2	15.5 ± 17.2	16.4 ± 9.4	24.5 ± 29.0
Stress Index Lying	98 ± 85	638 ± 654	350 ± 205	151 ± 252	75 ± 80
Stress Index Standing	168 ± 116	1111 ± 666	734 ± 632	561 ± 437	607 ± 578
HF lying	2920 ± 4403	210 ± 201	383 ± 517	2151 ± 4315	3798 ± 5574
HF standing	949 ± 1222	104 ± 170	212 ± 452	123 ± 126	813 ± 1981
LF lying	1518 ± 2795	226 ± 153	449 ± 364	1280 ± 1585	1021 ± 569
LF standing	1331 ± 1115	527 ± 413	242 ± 201	570 ± 632	1047 ± 1494
VLF lying	1553 ± 2182	639 ± 818	457 ± 368	1094 ± 1867	2103 ± 2483
VLF standing	1299± 1506	242 ± 201	455 ± 421	451 ± 440	864 ± 1245
TP lying	5819 ± 6203	876 ± 1164	1290 ± 349	4509 ± 6300	6435 ± 5318
TP standing	3579 ± 3012	349 ± 1132	1194 ± 1132	1145 ± 980	2726 ± 3910
LF/HF Lying	0.97 ± 1.10	3.08 ± 2.8	2.61 ± 2.67	1.12 ± 0.82	0.71 ± 0.51
LF/HF Standing	2.54 ± 1.95	4.85 ± 4.8	5.54 ± 2.79	7.66 ± 4.89	5.79 ± 6.94

HR: Heart Rate; FS: Fractional Shortening of the left ventricle; LVIMP: The Left Ventricular Index of Myocardial Performance; RMSSD: The square root of the mean of the sum of the squares of differences between adjacent NN intervals; TP: Total Power; HF: High-frequency power; LF: Low-frequency power; HF/LF: Ratio HF to LF; VLF: Very low-frequency power. Unpaired t-test within the patient groups “Inappropriate Sinus Tachycardia” and “Postural Orthostatic Tachycardia”: * *p*-value < 0.05; ** *p*-value < 0.01.

**Table 2 children-10-00316-t002:** The effect of the treatment on heart rate regulation in the active standing test.

	**Propranolol + O3-FA (*n* = 18)**		**Ivabradine + O3-FA** **(*n* = 11)**		**O3-FA without Pharmacotherapy (*n* = 7)**	
	**Postural Orthostatic Tachycardia Syndrome**					
**HR Increase**	42.2 ± 15.9	27.2 ± 17.4 ***	37.5 ± 16.4	23.6 ± 8.12 *	44.0 ± 11.9	25.6 ± 8.4 *
**HR Lying**	85.3 ± 21.4	75.8 ± 11.8	80.6 ± 17.9	77.9 ± 14.5	82.2 ± 17.6	84.3 ± 13.9
**HR Standing**	127.6 ± 18.1	101.9 ± 18.9 ***	118.2 ± 15.0	101.4 ± 16.9 **	126.2 ± 11.4	109.9 ± 16.9 *
**RMSSD Lying**	44.2 ± 32.7	67.0 ± 59.4	52.0 ± 17.0	44.0 ± 6.0	87.6 ± 35.0	54.6 ± 12.4
**RMSSD Standing**	11.3 ± 9.8	32.7 ± 34.4 *	17.5 ± 3.8	24.5 ±9.5	11.8 ± 3.2	27.2 ± 12.6
**Stress Index Lying**	314 ± 521	122 ± 112	232 ± 73	185 ± 38	192 ± 182	146 ± 111
**Stress Index Standing**	989 ± 719	401 ± 374 **	609 ± 123	548 ± 197	855 ± 160	561 ± 217
	**Inappropriate Sinus Tachycardia**					
	**Propranolol + O3-FA (*n* = 14)**		**Ivabradine + O3-FA** **(*n* = 6)**		**O3-FA without Pharmacotherapy (*n* = 11)**	
**HR Increase**	29.3 ± 16.0	19.4 ± 15.6	25.3 ± 11.6	21.2 ± 15.7	25.1 ± 12.5	23.5 ± 9.7
**HR Lying**	102.7 ± 20.8	81.6 ± 10.1 **	103.3 ± 12.2	84.2 ± 8.4 ***	96.4 ± 8.6	88.6 ± 7.9 *
**HR Standing**	132.1 ± 16.5	101.0 ± 18.8 ***	128.6 ± 12.2	105.4 ± 14.6 **	121.5 ± 11.9	112.1 ± 14.9 *
**RMSSD Lying**	26.7 ± 22.6	63.8 ± 54.5 **	15.0 ± 7.9	26.8 ± 14.3 *	28.3 ± 14.3	40.6 ± 21.1
**RMSSD Standing**	10.7 ± 7.9	46.3 ± 39.1 **	11.2 ± 8.6	13.8 ±8.1	19.1 ± 18.1	17.9 ± 10.4
**Stress Index Lying**	573 ± 729	123 ± 81 *	667 ± 633	272 ± 179	313 ± 186	207 ± 159
**Stress Index Standing**	1126 ± 759	403 ± 406 **	1083 ± 801	617 ± 353	613 ± 384	753 ± 1262

HR: Heart Rate; RMSSD: The square root of the mean of the sum of the squares of differences between adjacent NN intervals; O3-FA: omega-3 fatty acid supplementation. Paired t-test between baseline and therapy: * *p*-value < 0.05; b ** *p*-value < 0.01; *** *p*-value < 0.001.

## Data Availability

Anonymized data are available from the author.

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
