# Peer review of "Therapeutic Approaches to Dysautonomia in Childhood, with a Special Focus on Long COVID"

_children, 2023, doi:10.3390/children10020316_

Round 1
Reviewer 1 Report
The work with the title " Therapeutic approaches to Dysautonomia in Childhood with a special interest on Post-Acute Sequelae of Coronavirus 2019 disease and/or vaccination” is important and interesting.
The paper with the title "Therapeutic approaches to Dysautonomia in Childhood with a special interest on Post-Acute Sequelae of Coronavirus 2019 disease and/or vaccination" addresses an interesting topic, but the scientific research on this topic being neglected.
However, there are several questions:
In the introduction you write :: ,, The current publication is a retrospective analysis of our clinical routine data that compare heart rate analysis from 28 adolescents who develop an Inappropriate Sinus Tachycardia (IST, N=13) or a Postural Orthostatic Tachycardia Syndrome (POTS, N=15) after COVID-19 disease and/or vaccination with 64 adolescents from our data base who developed dysautonomia due to psychosomatic diseases prior to the COVID-19 pandemic.”
At the Material and Method you write: ,, We retrospectively analyse the data of 92 consecutive children with the diagnosis of an Inappropriate Sinus Tachycardia (IST, N=33, age 15.0±2.6 years) or a Postural Orthostatic Tachycardia Syndrome (POTS, N=59, age 14.8±2.4 years) of the pediatric department…”
In discussions you write:: ,,To date we found 28 adolescents who develop Postural Orthostatic Tachycardia Syndrome (N=15) or Inappropriate Sinus Tachycardia (N=13) after Covid-19 disease and/or vaccination (15 of 28 patients with Covid-19 related dysautonomia are vaccinated).
Am I supposed to understand that there are 13 teenagers with adequate sinus tachycardia or 33?! Can you comment?
At 2. 4 Statistics you write:,, The study population was divided into two diagnosis groups divided into Covid-19/vaccination related patients and patients prior to the pandemic and one healthy control group published and measured prior to the pandemic. "
In Table 1, 3 columns appear, although you write that there are only 2 groups. How do you comment?
You have conclusions only in the abstract, not in the text.
The references are not correctly included in the text, according to the template, for example: ,, For a better understanding of our therapeutic approaches, we recommend the current reviews upon Inappropriate Sinus Tachycardia12 and Postural Orthostatic Tachycardia Syndrome 13 who updated our knowledge with respect to autoantibodies after COVID -19 that interfere with receptors of the autonomic nervous system."
It is a small number of references, (13)
Dear authors, this is an interesting topic and I appreciate your effort.
My comments are only intended to make the paper better.
Good luck !
Reviewer 2 Report
This research article is relevant to the journal. The followings are my observations and suggestions while reviewing the paper.
1) Sample size is very small only 92. More variety of sample should be examine so that we could have more holistic view.
2) Methodology development could be improve with more elaboration.
3) Visualization is a big part for demonstrating the results which are not promising in this case and require modifications and more diagrams.
Reviewer 3 Report
In this study, 28 adolescents who develop an IST (N=13) and POTS (N=15) after COVID-19 disease and/or vaccination were compared with 64 adolescents who developed dysautonomia due to psycho-somatic diseases before the COVID-19 pandemic. It is relevant given that PASC is an increasing health problem in adults and children and the impact of SARS-CoV-2 vaccination on PASC is still controversial. Besides, the author tests the effect of propranolol (low dose up to 20-20-0mg, N=32) or ivabradine 5-5-0 mg (N=17) as treatment, both supplemented with Omega-3-Fatty Acid on heart rate and heart rate variability in this population.
I give my comments that allow this study to be published:
Major comments
- In the introduction, the author says “We observe a further increase of such non-specific symptoms in children and adults during the COVID-19 pandemic that may be related to the impact of SARS-CoV-2 infections on the autonomic nervous system”, Did you observe this? Reference 2 is a review from Bisaccia G., et al 2021.
- In “Material and methods”, there are severe inconsistencies, in the resume are mentioned 28 patients, but in methodology, there are no inclusion criteria, and in the results, the tables have other numbers. It must be consistent. The author says: “The methodology of the active standing test and the healthy control data of 47 children with a comparable mean age of 14.2 ±3.8 years are published”. Postural Orthostatic Tachycardia Syndrome must be defined and only referenced by Buchhorn and Buchhorn, 2020. What were the criteria to define Inappropriate Sinus Tachycardia? The statements that are referenced with 7, 8, and 9 must be included in the introduction section.
- The results, should not start with a table, this section is recommended to start with the description of the results, and when the table is referenced, it should be placed after the end of the sentence. In table 1, all abbreviations must be described in the table footer. The analysis must be included the treatment with only Propranolol and Ivabradine, followed by statistical analysis (table 2). This information allows for discriminating between treatments and supplementation. Figures 1 and 2 are not necessary, the author must describe in results. If the author decides to include figures 3 and 4, he should be included in the results section and described in the manuscript. I suggest performing the analysis between patients with COVID-19 and Vaccinated patients against SARS-CoV-2. The figure captions should contain more information for a better understanding of the graphs without having to go to the results.
- The discussion seems to result; the authors must carry out a correct discussion of their results. I do not understand why there are figures in the discussion.
Furthermore, it is necessary to include a section on the strengths and disadvantages of your study and a section on conclusions.
Round 2
Reviewer 2 Report
This journal paper already undergoes review process and improves a lot in the revised version.
Reviewer 3 Report
I suggest formatting the manuscript for publication